# LMCC-MBC: Metric-Constrained Model-Based Clustering with Wasserstein-2 Distance of Gaussian Markov Random Fields

## Abstract

A wide range of temporal (1D) and spatial (2D) data analysis problems can be formulated as model-based clustering problems given metric constraints. For example, subsequence clustering of multivariate time series is constrained by 1D temporal continuity, while urban functional area identification is constrained by the spatial proximity in the 2D space. Existing works model such metric constraints independent of the model estimation process, failing to leverage the correlation between adjacent estimated models and their locations in the metric space. To solve this problem we propose a novel metric-constrained model-based clustering algorithm LMCC-MBC that softly requires the Wasserstein-2 distance between estimated model parameters (such as those of Gaussian Markov Random Fields) to be a locally monotonic continuous function of the metric distance. We theoretically prove that satisfaction of this requirement guarantees intra-cluster cohesion and inter-cluster separation. Moreover, without explicitly optimizing log-likelihood LMCC-MBC avoids the expensive EM-step that is needed by previous approaches (e.g., TICC and STICC), and enables faster and more stable clustering. Experiments on both 1D and 2D synthetic as well as real-world datasets demonstrate that our algorithm successfully captures the latent correlation between the estimated models and the metric constraints, and outperforms strong baselines by a margin up to 14.3% in ARI (Adjusted Rand Index) and 32.1% in NMI (Normalized Mutual Information).

## 1 Introduction

Clustering is one of the most fundamental problems in unsupervised learning, which deals with the data partitioning when ground-truth labels are unknown (Xu & Tian, 2015). Most existing clustering algorithms only consider the similarity among observations in the feature space. However, in real-world applications, additional *metric constraints* (e.g., temporal continuity and geospatial proximity) often exist, especially in temporal and spatial data mining (Birant & Kut, 2007; Belhadi et al., 2020; Hu et al., 2015). In other words, observations do not only have features, but are also assigned positions; observations that are put in the same cluster should not only be similar in terms of features but their positions in the metric space (e.g., time points or geographic locations) should also satisfy some constraints. Metric constraints can be generalized to even higher dimensions as long as a meaningful distance measure is defined. This kind of clustering problems is commonly known as *metric-constrained clustering* (Veldt et al., 2019).

In metric-constrained clustering, there are fundamentally two separate spaces: the *feature space* where we compute the similarity among observations' features and perform clustering, and the *metric-constraint space* which imposes additional constraints on the clustering results. It is **NOT** a trivial question to design a larger composite space from these two spaces because the two spaces may have completely different metrics. For example, concatenating a word embedding with a geo-coordinate makes learning good similarity functions difficult because the former uses cosine distance while the latter uses Euclidean/geodesic distance, whose values can not be directly compared.

The current SOTA metric-constrained clustering, namely TICC (Hallac et al., 2017) and STICC (Kang et al., 2022), consider both spaces by combining model-based clustering with a soft metric penalty.

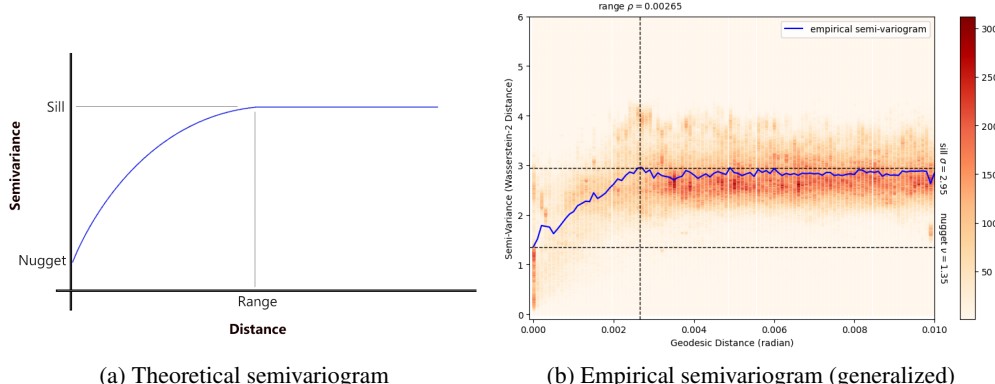

(a) Theoretical semivariogram  (b) Empirical semivariogram (generalized)

Figure 1: Comparison between (a) the classic theoretical semivariogram and (b) the generalized empirical model-based semivariogram computed on the iNaturalist-2018 dataset (North America only). Color in (b) denotes the number of observation pairs falling into each 0.0001(geodesic) × 0.01(Wasserstein-2) bin. We can see strong spatial autocorrelation from the figure.

These methods treat the feature vectors of observations in a given cluster as realizations of a common underlying (multi-variate) parametric stochastic process, referred to as the *underlying model*, and its parameters are statistically estimated from the observations. Then, the negative log-likelihood of an observation belonging to this cluster can be computed. A constant penalty is added to the negative log-likelihood if the observation is too far away the cluster in the metric-constraint space. An observation is accepted as a cluster member if the penalized negative log-likelihood is sufficiently small.

The advantage of this strategy is three-fold. Firstly, model-based clustering is by nature more robust to noise and outliers than algorithms based on pairwise feature similarity (Wang et al., 2016). Secondly, estimated underlying models provide better interpretability (Hallac et al., 2017). Finally, the magnitude of penalty can be tuned to adjust the emphasis on metric constraints, preferable to methods that enforce metric constraints as hard rules, such as ST-DBSCAN (Birant & Kut, 2007).

However, there is still room for major improvements. The most critical weakness of TICC/STICC is that they ignore the effects of metric autocorrelation, e.g., temporal/spatial autocorrelation (Goodchild, 1987; Anselin, 1988; Fortin et al., 2002; Gubner, 2006) when applying the metric penalty. In metric-constrained clustering, temporal/spatial autocorrelation effectively asserts that within a cluster, feature vectors observed at temporally/spatially distant positions should have higher variance than temporally/spatially adjacent ones, even though they both conform with the same underlying model. Intuitively that means the threshold we set for accepting a distant observation should be more tolerant than accepting an adjacent one, or equivalently, the metric penalty should be a decreasing function of distance instead of a constant. The intuition is that if two observations have identical likelihood of belonging to a cluster, the more distant one should be preferred, because autocorrelation implies that it would have higher likelihood if it were closer.

Besides, the way TICC/STICC implements the strategy incurs unwanted computational complexity and instability. Firstly, TICC/STICC uses expectation–maximization (EM) to optimize cluster assignments. This requires re-assigning observations and re-estimating underlying models every iteration, which is very time-consuming. Adding to that, the metric penalty designed as a binary function depending on nearest neighbors makes the optimization objective highly non-continuous and highly non-convex. As is admitted in the original papers (Hallac et al., 2017; Kang et al., 2022), this results in unstable convergence of the EM iterations. Finally, tuning the magnitude of the metric penalty is particularly difficult as a consequence of non-convexity and long, unstable convergence.

In this paper, we propose a novel model-based clustering method named LMCC-MBC (**L**ocally **M**onotonically and **C**ontinuously **C**onstrained **M**odel-**B**ased **C**lustering) that addresses the metric autocorrelation problem by designing a natural Wasserstein-2 distance-based multivariate generalization of the univariate semivariogram, which is widely used in geostatistics (Isaaks & Srivastava,

1989). We also demonstrate that the techniques developed can naturally mitigate the computational complexity and convergence instability problems of TICC/STICC.

In our method, we first fit a Gaussian Markov Random Field (GMRF) model for each observation $X_i$ and its metric neighbors. Then, instead of optimizing the similarity in the raw feature space, we choose to optimize the model distance between GMRFs measured by Wasserstein-2 distance. This allows us to incorporate metric autocorrelation directly into the optimization objective. Finally, we transform the autocorrelation-adjusted optimization objective into a plain clustering objective in the weighted distance space, and design an efficient algorithm to solve it.

To summarize, the **major contributions** of this paper are:
- We propose a generalized, mathematically clear formulation of metric constraint and metric-constrained clustering. It summarizes existing temporal/spatial metric-constrained clustering problems under a unified framework.
- We identify the problem of ignoring metric autocorrelation in existing SOTA methods.
- We propose a generalized, Wasserstein-2 distance-based definition of semivariogram that allows modeling multivariate metric autocorrelation. We present a solid mathematical proof of the soundness of this generalization and demonstrate its power in revealing cluster structures.
- We study a specified metric-constrained clustering problem that uses GMRFs and is widely applicable in many temporal/spatial clustering situations. We develop a novel method, LMCC-MBC, based on the weighted distance of Wasserstein-2 distance and constraint metric distance that combines the advantages of model-based and density-based clustering and solves the metric-constrained clustering problem under LMCC efficiently with solid theoretical proofs.
- We compare our method with existing works comprehensively on various 1D and 2D synthetic and real-world datasets. We demonstrate that our method outperforms the baselines both in clustering quality and computational efficiency.

## 2 RELATED WORKS

While the complete definition of clustering has not yet come to an agreement, three principles in general apply (Jain & Dubes, 1988): (1) **Intra-Cluster Cohesion**: observations, in the same cluster, must be as similar as possible; (2) **Inter-Cluster Separation**: observations, in different clusters, must be as different as possible; (3) **Interpretability**: measurement for similarity and dissimilarity must be clear and have practical meanings. Following these principles, the two key components of a clustering algorithm are the similarity/dissimilarity measurement and the algorithm that optimizes intra-cluster cohesion and inter-cluster separation. There is no cure-all clustering algorithm. Different data structures require different similarity/dissimilarity measurements (e.g. cosine distance, Euclidean distance, graph distance) and different optimization algorithms (e.g. hierarchical, iterative, estimation-maximization-based), resulting in a variety of clustering algorithms such as partition-based clustering, hierarchical clustering, density-based clustering, model-based clustering, etc. Please refer to Appendix 7.1 for a formal definition of clustering objectives.

Clustering temporal subsequences and spatial subregions is a well-studied sub-field. Some works treat temporal/spatial information as indices, such as dynamic time warping (Begum et al., 2015; Keogh, 2002; Keogh & Pazzani, 2000; Rakthanmanon et al., 2012), time point clustering (Gionis & Mannila, 2003; Zolhavarieh et al., 2014) and geo-tagged images (Liu et al., 2018; Mai et al., 2018), and some works cluster the spatio-temporal trajectories directly (Belhadi et al., 2020; Kisilevich et al., 2010). We are mostly interested in the first case, i.e., clustering temporally/geospatially referenced observations. However, these methods generally perform clustering based on feature similarity, which can be very problematic, or even unreliable (Keogh et al., 2003), because it only considers the structure of the feature space, ignoring that the observations are also distributed over time and space.

To address this problem, two main strategies are explored in previous works. The first strategy is to enforce metric constraints as hard rules. For example, in ST-DBSCAN (Birant & Kut, 2007), only temporally dense observations are considered candidates for core observations. The second strategy is to add a soft metric penalty to the clustering optimization objective. TICC (Hallac et al., 2017) is the first work to introduce a Markov Random Field to model temporal dependency structures of subsequences together with a soft temporal penalty. STICC (Kang et al., 2022), following this work, modified the algorithm to suit 2-dimensional spatial subregion clustering. They are both model-based clustering algorithms, like ARMA (Xiong & Yeung, 2004), GMM (Fraley & Raftery, 2006) and Hidden Markov Models (Smyth, 1996). They achieve SOTA performance.

However, TICC and STICC have their own drawbacks. To better understand this, we need to see the formulation of their optimization objective:

$$\arg\min_{\Theta, \mathcal{C}} \sum_{k=1..K} \left[ \left\| \lambda \circ \theta_{C_k} \right\|_1 + \sum_{X_i \in C_k} \left( -ll(X_i, \theta_{C_k}) + \beta \mathbb{1}\{\tilde{X}_i \notin C_k\} \right) \right] \tag{1}$$

Here $X_i$ is the observation we need to assign to a cluster, and $\tilde{X}_i$ is the nearest neighbor of $X_i$ in the metric space. $\theta_{C_k}$ is the estimated model parameters for cluster $k$, $C_k$ is the set of observations of cluster $k$, $-ll(X_i, \theta_{C_k})$ is the negative log-likelihood of observation $X_i$ given $\theta_{C_k}$, and $\beta \mathbb{1}\{\tilde{X}_i \notin C_k\}$ is the soft metric penalty. $\lambda$ is an L-1 normalization hyperparameter irrelevant to our discussion. From the formulation we can see the metric penalty term only depends on metric distance, failing to model the metric autocorrelation.

Other minor issues of TICC/STICC also exist. Firstly, as a model-based approach, they require presetting cluster numbers, which in most real-world applications is impossible. Secondly, they use temporal/spatial penalty terms inside of an E-M algorithm, which makes the optimization objective non-convex and intractable (Hallac et al., 2017). We find that Gaussian Markov Random Fields (GMRF) (Rue & Held, 2005) and Wasserstein-2 distance (Gibbs & Su, 2002) can be combined to solve these two issues.

A Gaussian Markov Random Field (GMRF) is a special case of the general Markov Random Field (MRF) (Wang et al., 2013), which additionally requires the joint and marginal distributions of variables to be Gaussian. Using GMRFs introduces several advantages. The first advantage is high computational efficiency. A (centered) GMRF can be efficiently represented and fitted as a sparse covariance matrix, through Graphical LASSO[1] (Friedman et al., 2007). Secondly, a GMRF can provide interpretable insights into variable correlations. Finally, a GMRF can be used to properly model continuous data in a wide range of situations (Rue & Tjelmeland, 2002; Hartman & Hössjer, 2008). For example, in spatial data mining, many commonly used real-valued features, such as check-in numbers (McKenzie et al., 2015; Janowicz et al., 2019), traffic volume (Liu et al., 2017; Cai et al., 2020), customer rating (Gao et al., 2017), and real-estate pricing (Law et al., 2019; Kang et al., 2021), can be treated as normal distributions after standardization. In addition to that, the covariance representation of a GMRF can be easily extended into a Toeplitz matrix that models inter-observation dependency, which is very important in understanding the interactions across time (Hallac et al., 2017) and space (Kang et al., 2022). Due to the above advantages of GMRF, we choose GMRFs as the parametrization of the underlying models in our method. Furthermore, the other important component of our method, the Wasserstein-2 distance (Gibbs & Su, 2002), works best with GMRFs. It is mathematically proved that the Wasserstein-2 distance has a closed-form solution on GMRF models, which ensures the efficiency and stability of our method.

## 3 PROBLEM FORMULATION

### 3.1 METRIC-CONSTRAINED CLUSTERING

Given a dataset $\mathcal{D}$ of $N$ observations $\{X_i\}_{i=1}^N$ (e.g., points of interest in an urban area, sensor measurements at different time points, etc), we need to assign the index of each $X_i$ to a set $C_k$, i.e. cluster $k$. The set of all clusters $\mathcal{C} = \{C_k\}_{k=1}^K$ is a *cluster assignment* or a *clustering*. $K$ is called the number of clusters, either predefined or inferred from data.

Each observation $X_i = (\mathbf{f}_i, \mathbf{p}_i)$ is a tuple of two vectors: $\mathbf{f}_i$ is a $\mathrm{d}_F$-dimensional feature vector (e.g., attributes of a POI) in a feature space $F$, while $\mathbf{p}_i$ is a $\mathrm{d}_M$-dimensional position vector (e.g., geo-coordinates of this POI) in a metric space $(M, d_c)$ (e.g., Earth surface with geodesic distance), where $d_c$ is a predefined metric. With a dissimilarity measurement $d_f(\cdot, \cdot)$ in the feature space, e.g., cosine distance, a classic clustering problem without metric constraints $\hat{\mathcal{C}}_K = \arg\min_{\mathcal{C}} \mathcal{L}(\mathcal{C})$ is to minimize the objective:

$$\mathcal{L}(\mathcal{C}) = \sum_{\{C_k \in \mathcal{C}\}} \sum_{\{i,j \in C_k\}} d_f(\mathbf{f}_i, \mathbf{f}_j) - \alpha \sum_{\{C_k, C_l \in \mathcal{C}, k \neq l\}} \sum_{\{i \in C_k, j \in C_l\}} d_f(\mathbf{f}_i, \mathbf{f}_j) \tag{2}$$

where the first term is the intra-cluster cohesion objective and the second is the inter-cluster separation objective. $\alpha$ is a hyperparameter balancing cohesion and separation.

---

[1]https://scikit-learn.org/stable/modules/generated/sklearn.covariance.graphical_lasso.html

A *metric constraint* is an additional objective $\mathcal{L}^{\text{mc}}$ that assigns penalty based on metric distance and feature similarity. A *metric-constrained clustering* problem is to find an optimal cluster assignment that minimizes a multi-objective $\hat{\mathcal{C}}_K^{\text{cons}} = \arg\min_{\mathcal{C}} \left[ \mathcal{L}(\mathcal{C}) + \beta \mathcal{L}^{\text{mc}}(\mathcal{C}) \right]$ where

$$\mathcal{L}^{\text{mc}}(\mathcal{C}) = \sum_{\{C_k \in \mathcal{C}\}} \sum_{\{i,j \in C_k\}} r(d_f(\mathbf{f}_i, \mathbf{f}_j), d_c(\mathbf{p}_i, \mathbf{p}_j)) \tag{3}$$

$r$ is a function of the metric distance (and potentially of the feature similarity, too) called the *metric penalty function*, designed to properly enforce the metric constraints. $\beta$ is a hyperparameter that determines how soft the constraints are. For example, in ST-DBSCAN (Birant & Kut, 2007), temporal continuity is the metric constraint. Conceptually its $r(d_m(\mathbf{f}_i, \mathbf{f}_j), d_c(\mathbf{p}_i, \mathbf{p}_j)) = \mathbb{1}\{d_c(\mathbf{p}_i, \mathbf{p}_j) > \epsilon_t\}$, $\epsilon_t$ being the preset radius of temporal neighborhood, and $\beta = \infty$. It effectively means that cluster assignments with temporal discontinuity are hard eliminated.

## 3.2 Metric-Constrained Model-Based Clustering

Metric-constrained model-based (MCMB) clustering is a special case of metric-constrained clustering. As is previously discussed, model-based clustering views the feature vector $\mathbf{f}_i$ of observation $X_i$ as a random sample drawn from a parametric distribution $\mathcal{M}(\theta_i)$. We say $\mathcal{M}(\theta_i)$ is the underlying model of $X_i$ and $\theta_i$ is a specification of the parameters. The family of distribution (e.g. Gaussian, Poisson, etc.) and exact parameterization (e.g. mean+covariance or mean+inverse covariance) of $\mathcal{M}(\theta_i)$ is chosen a priori based on domain knowledge and computational considerations. In MCMB clustering, the feature dissimilarity measure $d_f(\mathbf{f}_i, \mathbf{f}_j)$ is replaced with $d_m(i,j) = d_m(\mathbf{f}_i, \mathbf{f}_j, \theta_i, \theta_j; \mathcal{M})$, named as *model-based dissimilarity*, or simply *model distance*.

In summary, MCMB clustering can be formulated as minimizing the multi-objective MCMB loss[2]

$$\mathcal{L}^{\text{mcmb}}(\mathcal{C}) = \sum_{\{C_k \in \mathcal{C}\}} \sum_{\{i,j \in C_k\}} [d_m^2(i,j) + \beta r(i,j)] - \alpha \sum_{\{C_k, C_l \in \mathcal{C}, k \neq l\}} \sum_{\{i \in C_k, j \in C_l\}} d_m^2(i,j) \tag{4}$$

In most scenarios, $\alpha$ is rather small and the second term can be omitted compared to the first term (because people usually emphasize more on intra-cluster cohesion). For example, in TICC/STICC (Hallac et al., 2017; Kang et al., 2022), the authors take $\alpha = 0$.

## 4 Methodology

As shown in Equation 17, three key pieces must be specified a priori for a MCMB clustering problem: 1) the family of distribution and exact parametrization of the underlying model $\mathcal{M}(\theta)$; 2) the exact expression of the model distance function $d_m$; 3) the exact expression of the metric penalty function $r$. Our proposed method, LMCC-MBC, is a novel, integral solution in that all its three pieces are naturally developed from solving the metric autocorrelation problem. In this section, we will deduce step by step how we start from classic spatial analysis theories and arrive at the combination of Gaussian Markov Random Fields, Wasserstein-2 distance and the LMCC penalty function.

### 4.1 Generalized Model-based semivariogram

In the introduction we have argued for the importance of metric autocorrelation in metric-constrained clustering. It is the core research problem of this paper. The first step is to appropriately quantify it.

While an abundance of statistics for autocorrelation tests are developed in classic temporal and spatial analysis, such as Durbin–Watson statistic (Durbin & Watson, 1950; 1951) and Moran's I (Moran, 1950), the semivariogram (Matheron, 1963) fits our end best. This is because the theoretical semivariogram, denoted as $\gamma(\mathbf{p}_i, \mathbf{p}_j)$, is a function describing the degree of spatial dependence of a spatial random field or stochastic process, which is literally the fundamental assumption of model-based clustering. Figure 1a is an illustration of the theoretical semivariogram. We can see

---

[2]For the rest of the paper, we abbreviate the notations by omitting the arguments and only keeping the indices. For example, $d_m(\mathbf{f}_i, \mathbf{f}_j, \theta_i, \theta_j; \mathcal{M})$ is written in short as $d_m(i,j)$, $d_c(\mathbf{p}_i, \mathbf{p}_j)$ as $d_c(i,j)$, $r(d_m(i,j), d_c(i,j))$ as $r(i,j)$, respectively.

in the beginning, the curve rises as distance increases, which is the sign of the presence of spatial autocorrelation; then it gradually flattens out, which is the sign of the disappearance of spatial autocorrelation. Three key concepts lie in this figure: the range (the distance beyond which spatial autocorrelation disappears), the sill (the semi-variance when spatial autocorrelation disappears) and the nugget (the semi-variance when distance is almost zero; this is considered as the intrinsic variance of the stochastic process).

In practice, given a dataset (which is a sample generated from the spatial stochastic process) of $N$ univariate observed variables $\{z_1, \cdots z_N\}$ together with their spatial positions $\{\mathbf{p}_1, \cdots \mathbf{p}_N\}$, there are $N^2$ pairs of variables $(z_i, z_j)$ and their corresponding pairs of spatial positions $(\mathbf{p}_i, \mathbf{p}_j)$. The empirical semivariogram is defined as

$$\hat{\gamma}(h \pm \epsilon) := \frac{1}{2|N(h \pm \epsilon)|} \Sigma_{\{(\mathbf{p}_i, \mathbf{p}_j) \in N(h \pm \epsilon)\}} |z_i - z_j|^2 \tag{5}$$

where $N(h \pm \epsilon) := \{(\mathbf{p}_i, \mathbf{p}_j) | h - \epsilon \le d_c(\mathbf{p}_i, \mathbf{p}_j) \le h + \epsilon\}$. This is essentially the half empirical variance of all pairs whose spatial distance falls into the same bin centered at $h$ of width $2\epsilon$.

Semivariogram can also be applied to metric spaces other than 2-dimensional or 3-dimensional geospatial space, such as temporal space, spatio-temporal space and even multi-dimensional, non-Euclidean spaces (Nguyen et al., 2014). However, it can not be applied to multi-variate observations. This is because the concepts of range, sill and nugget are defined as turning/intercepting points of the function; if $\gamma$ is multi-variate, the three core concepts are not well-defined.

We propose a novel multivariate generalization of the classic semivariogram, called *generalized model-based semivariogram*. Unlike existing works such as Abzalov (2016), which modifies the definitions of range, sill and nugget analogously to simultaneous confidence intervals, we derive a natural generalization of the variance. As in model-based clustering, every observed feature vector $\mathbf{f}_i$ has an underlying model $\mathcal{M}(\theta_i)$. Though the difference between the feature vectors is multivariate, the difference between the underlying models is univariate. Then the empirical generalized model-based semivariogram $\hat{\gamma}_m$ can be defined as

$$\hat{\gamma}_m(h \pm \epsilon) := \frac{1}{2|N(h \pm \epsilon)|} \Sigma_{\{(\mathbf{p}_i, \mathbf{p}_j) \in N(h \pm \epsilon)\}} d_m^2(i, j) \tag{6}$$

The soundness of this generalized definition is theoretically supported by goodness-of-fit tests. Detailed discussion can be found in Appendix Section 7.4. It can also be easily verified on real-world datasets. Figure 1b is the empirical generalized model-based semivariogram computed on a large geo-tagged image dataset iNaturalist-2018 (Cui et al., 2018). We use the top 16 PCA components of the pretrained embedding of each image as the feature vector, and the metric constraint is the geodesic distance (i.e., great circle distance) between the geo-tags. For each image, we use its 30-nearest neighbors to estimate a Gaussian Markov Random Field (GMRF) as the underlying model. We use Wasserstein-2 distance as $d_m$. Then we compute the generalized semivariogram using Equation 6. We can see clearly the empirical semivariogram conforms very well with the theory.

We wish to point out that the choice of Wasserstein-2 distance and GMRF is not heuristic. In fact, Wasserstein-2 distance is the only feasible choice of model distance that theoretically guarantees the generalized model-based semivariogram is compatible with the classic definition. Consequently, GMRF is chosen because it is the most computationally efficient model parameterization under Wasserstein-2 distance. The following section will justify this in detail.

## 4.2 WASSERSTEIN-2 DISTANCE AND GAUSSIAN MARKOV RANDOM FIELDS

There are various statistical metrics or quasi-metrics that can be used to quantify the similarity/dissimilarity between underlying models, i.e., distributions. Commonly used ones include divergence (such as KL-divergence), total variation, discrepancy and Wasserstein-2 distance (Gibbs & Su, 2002). Since we use model distance to define the generalized semivariogram, it is important that the choice of statistical metrics is compatible with the classic definition. Specifically, we wish to show that having small semi-variance in terms of model distance guarantees having small semi-variance in terms of feature difference. The weakest possible condition that satisfies this requirement is weak convergence, also known as convergence in distribution. Intuitively it says if a model weakly converges to another model, the observation generated from them will become statistically indistinguishable,

consequently having the indistinguishable semi-variance. Therefore, we need to find a statistical metric $d_m$ that metricizes weak convergence, i.e., $(d_m(i,j) \to 0) \Rightarrow (\mathcal{F}_i \xrightarrow{D} \mathcal{F}_j)$. Here $\mathcal{F}_i, \mathcal{F}_j$ are cumulative distribution functions parametrized by $\theta_i, \theta_j$ respectively, and $\xrightarrow{D}$ denotes convergence in distribution. Among all such metrizations, Lévy-Prokhorov metric and Wasserstein's distance are the two most important cases. Their definitions are as follows:

**Lévy-Prokhorov Metric**: given a separable metric space $(M, d)$ together with its Borel sigma algebra $\mathcal{B}(M)$, define the $\epsilon$-neighborhood of $A \subset M$ as $A^\epsilon := \{p \in M : \exists q \in A \text{ s.t. } d(p, q) < \epsilon\}$. Then the Lévy-Prokhorov metric $\pi$ of two probability measures $\mu, \nu$ is defined as

$$\pi(\mu, \nu) := \inf\{\epsilon > 0 : \mu(A) \le \nu(A^\epsilon) + \epsilon \text{ and } \nu(A) \le \mu(A^\epsilon) + \epsilon, \forall A \in \mathcal{B}(M)\} \tag{7}$$

**Wasserstein's Distance**: given a Radon metric space $(M, d)$, for $p \in [1, \infty)$, the Wasserstein-$p$ distance $W_p$ between two probability measures $\mu, \nu$ is defined as

$$W_p := (\inf_{\gamma \in \Gamma(\mu, \nu)} \mathbb{E}_{(x,y) \sim \gamma} d(x,y)^p)^{1/p} \tag{8}$$

where $\Gamma(\mu, \nu)$ is the set of all possible couplings of $\mu$ and $\nu$.

By Gibbs & Su (2002), Lévy-Prokhorov metric is the tightest bound of distance between two distributions, and the Wasserstein's distance is only looser up to a constant factor. Whereas the Lévy-Prokhorov Metric is in general not computable, the Wasserstein-2 distance between two Gaussian Markov Random Fields has a beautiful closed-form:

$$W_2^2(\theta_1, \theta_2) = d_2^2(\mu_1, \mu_2) + \text{Tr}(\Sigma_1 + \Sigma_2 - 2(\Sigma_1^{1/2} \Sigma_2 \Sigma_1^{1/2})^{1/2}) \tag{9}$$

Here $\mu_1, \mu_2$ are mean vectors and $\Sigma_1, \Sigma_2$ are covariance matrices. $\theta = (\mu, \Sigma)$. Tr is the trace of a matrix. Thus, the combination of Wasserstein-2 distance and Gaussian Markov Random Fields is essentially the only choice we have that both satisfies our need for modeling metric autocorrelation and comes with practical computability.

## 4.3 Locally Monotonic and Continuous Constraint (LMCC)

LMCC arises naturally from the generalized model-based semivariogram. As we have discussed in Appendix Section 7.4, $r(i, j)$ should be defined as

$$r(i, j) = \max(0, d_m^2(i, j) - (\gamma_m(d_c(i, j)) - \delta)) \tag{10}$$

Furthermore, for computational efficiency, we ignore the penalty term when $d_c(i, j) > \rho$ and the final form of $r(i, j)$ is defined as

$$r(i, j) = \left\{ \begin{array}{ll} \max(0, d_m^2(i, j) - \gamma_m(d_c(i, j)) + \delta), & d_c(i, j) \le \rho \\ 0, & d_c(i, j) > \rho \end{array} \right\} \tag{11}$$

where $\gamma_m$ is the theoretical generalized model-based semivariogram function fitted from the empirical variogram cloud (Müller, 1999), and $\rho$ is the range of the fitted semivariogram. Intuitively, LMCC penalizes observations whose semivariance exceeds the variogram. It is local (only effective within range $\rho$), monotonically decreasing and continuous (because the variogram is monotonically increasing and continuous within the range). See Appendix 7.4 for a detailed discussion on the theoretical reason and Appendix 7.7.4 for an ablation study.

$\delta$ is a critical hyperparameter called *shift* that needs to be tuned. The motivation of using this hyperparameter comes from Figure 2. By plotting the percentage of ground-truth intra-cluster pairs in each bin, we can see a clear boundary between the dark region (with higher percentage of intra-cluster pairs) and the light region (with lower percentage of intra-cluster pairs). If a pair of observations fall into the dark region, they are more likely to belong to the same cluster, and vice versa. This would be extremely useful information for clustering. However, we do not know where the boundary is if we do not know the ground-truth. Fortunately, we notice that if we vertically shift the semivariogram down for an appropriate distance $\delta$, it will partially overlap with the boundary. Then penalizing the observation pairs above the variogram becomes equivalent to penalizing the observation pair for falling into the region of low intra-cluster probability.

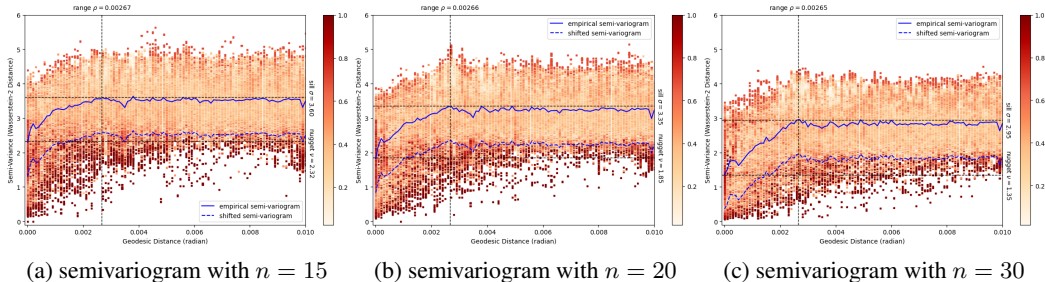

(a) semivariogram with $n = 15$     (b) semivariogram with $n = 20$     (c) semivariogram with $n = 30$

Figure 2: Empirical semivariogram under different hyperparameter settings. $n$ is the number of nearest neighbors used for fitting the GMRF models. The color represents the percentage of observation pairs that belong to the same ground-truth cluster in each $0.0001(\text{geodesic}) \times 0.01(\text{Wasserstein-2})$ bin.

---

**Algorithm 1:** Weighted Distance Algorithm for Approximately Solving the LMCC-Constrained Model-based Clustering Problem

---

**Input** : A dataset $\mathcal{D}$ of $N$ observations $\{X_i = (\mathbf{f}_i \in F, \mathbf{p}_i \in M)\}_{i=1}^N$. The distance function $d_m$. The metric penalty function $r$. The model fitting algorithm **GL**. The clustering algorithm **DB**. The number of neighbors $n$ used for model fitting. The metric-constraint strength $\beta$. The shift hyperparameter $\delta$.

**Output** : A clustering $\mathcal{C} = \{C_k\}_{k=1}^K$

1 for each observation $X_i \in \mathcal{D}$, find the set of $n$ neighboring observations $N_i$ in the metric-constraint space. Fit the model parameters $\theta_i \leftarrow \mathbf{GL}(N_i)$;

2 for each pair of observations, compute their model distance $d_m(i,j) \leftarrow d_m(\theta_i, \theta_j)$ and metric distance $d_c(i,j) \leftarrow d_c(\mathbf{p}_i, \mathbf{p}_j)$;

3 estimate a semivariogram $\gamma_m$ from $d_m(i,j)$ and $d_c(i,j)$; compute the range $\rho$;

4 compute the weighted distance matrix $M_{i,j}^w \leftarrow d_m(i,j) + \beta r(i,j)$;

5 run clustering algorithm $\mathcal{C} \leftarrow \mathbf{DB}(M^w)$;

6 **return** $\mathcal{C}$

---

### 4.4 WEIGHTED DISTANCE ALGORITHM

We implement our algorithm based on Equation 17, Equation 6 and Equation 11.

## 5 EXPERIMENTS

We performed extensive experiments on both 1D and 2D, both synthetic and real-world datasets. All values reported are obtained under the best hyperparamter setting found by grid search. The detailed experiment setup and evaluation metrics are in Appendix 7.3.

From Table 1 we can see in general, model-based algorithms handle spatio-temporally distributed data better than feature-based clustering algorithms. Our method (LMCC-w) outperforms the strong baselines (TICC and STICC) in all tasks. Besides performance improvement, LMCC-MBC is also more flexible and generally applicable. LMCC-MBC deals with arbitrary dimensions of constraint spaces and unknown cluster numbers without any modification, whereas TICC and STICC can only handle either 1D or 2D metric constraints and the cluster number must be preset.

Furthermore, comparing TICC/STICC/LMCC-MBC with their non-constrained version (i.e., TICC ($\beta = 0$) v.s. TICC, STICC ($\beta = 0$) v.s. STICC, or LMCC-wo v.s. LMCC-w), we can see that metric constraints do improve the clustering quality.

### 5.1 STABILITY, ROBUSTNESS AND EFFICIENCY

LMCC-MBC is robust and computationally stable in two ways. Firstly, it is a sequential algorithm. TICC/STICC, on the other hand, uses E-M iterations which may accumulate error. For example, we

Table 1: Performance on 1-D and 2-D real-world datasets. $d$ denotes the feature dimension, $c$ denotes the ground-truth cluster number and $N$ denotes the size of each dataset. **Bold** numbers and underlined numbers indicate the best and second best performances. (S)TICC means applying TICC to temporal datasets and STICC to spatial datasets. $\beta = 0$ means there is no temporal/spatial penalty term applied. NC means the algorithm does not converge. - means the method is not suitable for this dataset. LMCC-wo/LMCC-w represents LMCC model without/with metric constraints respectively.

| Model | Synthetic Datasets | | | | Real-world Datasets | | | | | | | | | | | | | |
| | Temporal | | Spatial | | Temporal | | | | | | | | Spatial | | | | | |
| | | | | | Pavement $d$=10, $c$=3 N=1,055 | | Vehicle $d$=7, $c$=5 N=16,641 | | Gesture $d$=3, $c$=8 N=704,970 | | Climate $d$=5, $c$=14 N=4,741 | | iNat2018 $d$=16, $c$=6 N=24,343 | | POI $d$=7, $c$=10 N=23,019 | | Landuse $d$=7, $c$=5 N=8964 | |
| | ARI | NMI | ARI | NMI | ARI | NMI | ARI | NMI | ARI | NMI | ARI | NMI | ARI | NMI | ARI | NMI | ARI | NMI |
| k-Means | 1.03 | 1.69 | 1.26 | 1.66 | 8.02 | 6.59 | 8.94 | 21.54 | 2.78 | 5.23 | 5.47 | 22.14 | 6.91 | 14.71 | 18.37 | 43.44 | 2.39 | 4.21 |
| DBSCAN | 2.44 | 2.50 | 3.69 | 5.38 | 15.25 | 18.75 | 33.67 | 41.83 | 1.18 | 2.07 | 3.61 | 17.89 | 34.91 | 34.69 | 15.03 | 39.29 | 11.91 | 7.19 |
| HDBSCAN | 0.90 | 0.61 | 1.00 | 1.39 | 7.10 | 11.66 | 37.51 | 41.64 | - | - | 11.52 | 28.01 | 7.65 | 17,92 | 20.78 | 62.55 | 1.00 | 7.64 |
| DTW | 2.52 | 2.13 | - | - | 17.13 | 17.55 | 8.11 | 23.35 | - | - | - | - | - | - | - | - | - | - |
| GMM | 7.82 | 9.54 | 9.26 | 10.35 | 28.05 | 28.74 | 57.87 | 58.78 | 2.44 | 4.15 | 19.06 | 34.97 | 21.72 | 35.91 | 16.38 | 42.96 | 2.86 | 4.61 |
| (S)TICC-$\beta = 0$ | 80.11 | 83.95 | 91.28 | 89.28 | 58.54 | 58.83 | 40.12 | 45.86 | 3.26 | 6.56 | 13.30 | 30.53 | NC | NC | 13.29 | 27.08 | 7.22 | 12.60 |
| (S)TICC | 84.88 | 86.13 | 91.84 | 89.85 | 62.27 | 61.89 | 50.53 | 53.68 | 12.20 | 23.20 | 17.62 | 37.29 | NC | NC | 17.12 | 39.85 | 11.04 | 15.35 |
| LMCC-wo | 86.38 | 84.56 | 87.34 | 84.74 | 76.10 | 74.36 | 63.31 | 58.60 | 8.12 | 33.60 | 16.63 | 36.73 | 21.90 | 36.47 | 30.45 | 66.23 | 12.91 | 28.72 |
| LMCC-w | **90.50** | **87.96** | **94.49** | **91.98** | **77.64** | **77.22** | **65.04** | **59.36** | **26.51** | **55.34** | **20.08** | **40.91** | **42.70** | **40.49** | **39.81** | **68.27** | **36.54** | **42.97** |

observe that if the initial cluster assignment is too imbalanced, TICC/STICC will self-enhancingly increase this imbalance until it fails to converge. Secondly, LMCC-MBC has only three hyperparameters to tune: the number of neighbors $n$; the penalty weight $\beta$ and the shift $\delta$. $\beta$ and $n$ are common hyperparameters that all model-based clustering algorithms share. Thus, only the shift $\delta$ is unique to our method. Furthermore, by comparing Figure 2a, 2b and 2c, we find that the key factors of the semivariogram (range, sill and nugget) remain relatively stable. This finding is critical because our method is heavily based on the reliable construction of the semivariogram.

Speaking of efficiency, as our method gets rid of the E-M process, the time consumption is consequently much less. Theoretically, our method only estimates the underlying model for each observation once throughout the entire algorithm, whereas TICC/STICC must re-estimate models in every iteration. Empirically, the execution speed of our method is 5 to 15 times faster than TICC/STICC. Moreover, the estimated underlying models can be saved, making hyperparameter tuning much easier than TICC/STICC. See Appendix 2 for further information.

## 6 CONCLUSION AND FUTURE WORKS

In this paper, we propose a novel method LMCC-MBC that combines the advantages of model-based clustering and distance-based clustering, by computing pairwise Wasserstein-2 distance between estimated model parameterizations for each observation. This method provides a unified solution to clustering problems with temporal/spatial/higher-dimensional metric constraints and achieves SOTA performance on both synthetic and real-world datasets. Moreover, LMCC-MBC is more computationally efficient and stable than the strongest baselines TICC and STICC. For future work, it is worth extending LMCC-MBC to non-Gaussian, general Markov Random Fields.

**Reproducibility Statement**

We have uploaded the code we use to reproduce the experiment results reported in our paper. Datasets we use, except the Climate dataset, are all publicly accessible and we point the readers to the sources in Appendix 7.3. All details of experiment setup, including how we generate synthetic datasets, how we define and compute the evaluation metrics are also clearly reported there.

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

# 7 APPENDIX

## 7.1 A GENERAL OBJECTIVE OF CLUSTERING

Conceptually, if we define measurements similarity $s(\cdot, \cdot)$ and dissimilarity $d(\cdot, \cdot)$ between two observations $X_i, X_j$, we can view a clustering problem as finding an optimal cluster assignment $\hat{C}_K$ that maximizes the objective:

$$\hat{C}_K = \arg\max_C \Big[ \sum_{\{C_k \in C\}} \sum_{\{i,j \in C_k\}} s(X_i, X_j) + \beta \sum_{\{C_k, C_l \in C, k \neq l\}} \sum_{\{i \in C_k, j \in C_l\}} d(X_i, X_j) \Big] \quad (12)$$

The first term corresponds to the intra-cluster cohesion principle and the second term corresponds to the inter-cluster separation principle. $\beta$ is a hyperparameter chosen to control how much we weigh these two terms, since the two objectives may compete. The choice of $s$ and $d$, in turn, corresponds to the interpretability principle. Usually we simply let $s(\cdot, \cdot) = -d(\cdot, \cdot)$, thus the objective becomes

$$\hat{C}_K = \arg\min_C \Big[ \sum_{\{C_k \in C\}} \sum_{\{i,j \in C_k\}} d(X_i, X_j) - \beta \sum_{\{C_k, C_l \in C, k \neq l\}} \sum_{\{i \in C_k, j \in C_l\}} d(X_i, X_j) \Big] \quad (13)$$

## 7.2 METRICIZATION OF WEAK CONVERGENCE

To read more detailed discussions of the metrization of probability convergence, see Gibbs & Su (2002) for a comprehensive summary. According to the same paper, two important propositions are worth notification: 1) the Lévy-Prokhorov metric is "precisely the minimum distance 'in probability' between random variables distributed according to $\mu$ and $\nu$", and 2) the Lévy-Prokhorov metric and the Wasserstein's distance satisfy the following quantitative relation:

$$\pi \leq W_p \leq (\text{diam}(\Omega) + 1)\pi \quad (14)$$

where $\text{diam}(\Omega) := \sup\{d(x, y) : x, y \in \Omega\}$ is the diameter of the sample space $\Omega$. These two propositions justify that though the Wasserstein's distance is not the tightest bound (i.e., the Prokhorov metric), it converges as fast up to a constant factor, so long as the metric space is bounded.

Since both the Lévy-Prokhorov metric and the Wasserstein's distance has guaranteed convergence, the choice of $d_m$ is mainly upon computational efficiency. Whereas both metrizations have no simple algorithms for computation in the general case, the Wasserstein-2 distance between two multi-variate Gaussian distributions has a neat closed-form formula in terms of mean vectors and covariance matrices. Gelbrich (1990) gives the formula of the squared Wasserstein-2 distance as follows:

$$W_2^2(\theta_1, \theta_2) = d_2^2(\mu_1, \mu_2) + \text{Tr}(\Sigma_1 + \Sigma_2 - 2(\Sigma_1^{1/2} \Sigma_2 \Sigma_1^{1/2})^{1/2}) \quad (15)$$

Here $\mu_1, \mu_2$ are mean vectors and $\Sigma_1, \Sigma_2$ are covariance matrices. $\theta = (\mu, \Sigma)$. Tr is the trace of a matrix. Thus, the combination of Wasserstein-2 distance and Gaussian Markov Random Fields is essentially the only choice we have that both satisfies our need for modeling metric autocorrelation and comes with practical computability.

By computing the pairwise Wasserstein-2 distance between the estimated models, we obtain a distance matrix. Any density-based clustering algorithms that support pre-computed distance matrix, such as DBSCAN(Ester et al., 1996), can be seamlessly applied without any modification. Since these clustering algorithms are designed to minimize intra-cluster distance and maximize inter-cluster distance, it follows immediately that the intra-cluster observations follow as similar as possible distributions whereas the inter-cluster observations follow as dissimilar as possible distributions, by the fact that the Wasserstein's distance is a metrization of weak convergence.

With this dimension reduction, we can finally transform the original metric-constrained model-based clustering problem in the product space $F \times M$ to a simpler problem in the product space $\mathcal{R} \times M$. Since $\mathcal{R}$ and $M$ are both metric spaces, density-based algorithms that are supported on product metric spaces such as ST-DBSCAN(Birant & Kut, 2007) can be then applied. However, these algorithms treat the two metric spaces independently without considering the correlation introduced by the metric constraint. In Section4.3 and Section4.4, we discuss how to address this issue.

### 7.3 EXPERIMENT SETUP

**Baseline Models**. We compare our method to both density-based and model-based clustering algorithms. See Table 1 for details. Among them, TICC(Hallac et al., 2017) can only deal with 1-dimensional constraint and STICC(Kang et al., 2022) can only deal with 2-dimensional constraint. Thus, the former will only be evaluated against 1-dimensional datasets and the latter only against 2-dimensional datasets. All other models that do not incorporate metric constraint information are evaluated on both 1-dimensional and 2-dimensional datasets.

**Evaluation Metrics of Clustering Quality**. For the fairness of comparison, we adopt the most commonly used ground-truth label based metrics, Adjusted Rand Index (ARI)[3](Hubert & Arabie, 1985) and Normalized Mutual Information (NMI)[4](Vinh et al., 2010). We use their implementation in `sklearn`(Pedregosa et al., 2011). We do not adopt the Macro-F1 metric that TICC(Hallac et al., 2017) uses because this metric is only well-defined when cluster number is fixed, while our method is density-based, which does not preset a cluster number.

**Synthetic Dataset** We generate 1-dimensional and 2-dimensional synthetic datasets following the LMCC assumption discussed in Section 4.3. The only hyperparameters we preset are cluster number $K$, feature dimension $D$, noise scale $\alpha$ and sample batch size $k$. All other hyperparameters such as sequence length, cluster size and so forth are completely randomly generated for the sake of fair comparison.

**1-Dimensional Synthetic Dataset**. We generate the 1-dimensional synthetic dataset following the LMCC assumption discussed in Section 4.3:

- Choose hyperparameters: cluster number $K$, feature dimension $D$, noise scale $\alpha$, sample size $k$.
- Randomly choose a subsequence number $N$.
- Randomly generate $K$ different $D \times D$ ground-truth covariance matrices $\{\Sigma_1, \Sigma_2 \cdots \Sigma_K\}$. It is required that the pairwise Wasserstein-2 distances should all be greater than 1.0. This is to make sure that observations of different clusters are statistically different.
- Generate a random list of $N$ ground-truth subsequence cluster labels $\{C_1, C_2 \cdots C_N\}, C_i \in \{0..K-1\}$, and a random list of $N$ subsequence lengths $\{L_1, L_2 \cdots L_N\}$.
- For each subsequence label $C_i$ and subsequence length $L_i$, generate $L_i$ perturbed covariance matrices $\{P_{i,1}, P_{i,2} \cdots P_{i,L_i}\}$ by adding Gaussian noise to $\Sigma_i$. Notice, in order to conform with the monotonic assumption, we add noise with noise scale $j\alpha$ as we generate $P_{i,j}$, and the maximum noise scale should be no larger than $10\%$ of the maximum entry in the ground-truth covariance matrix, in order to meet the continuous assumption.
- Sample $k$ $D$-dimensional feature vectors from each $P_{i,j}$ sequentially and concatenate them all together into a $k\Sigma_{i=1}^{N} L_i$ list, with each entry being a $D$-dimensional feature vector. The corresponding position list is simply $\{1, 2 \cdots k\Sigma_{i=1}^{N} L_i\}$. Pairing the feature list and the position list makes the dataset.

**2-Dimensional Synthetic Dataset**. We generate the 2-dimensional synthetic dataset, also following the LMCC assumption discussed in Section 4.3:

- Choose hyperparameters: cluster number $K$, feature dimension $D$, noise scale $\alpha$.
- Randomly choose a list of cluster sizes $\{N_1..N_K\}$.
- Randomly generate $K$ points $\{\mathbf{p}_1..\mathbf{p}_K\}$ on the $X-Y$ plane as the metric center of clusters. Randomly generate $K$ $2 \times 2$ covariance matrices $\{S_1..S_K\}$. For each $\mathbf{p}_i$, generate $N_i$ points $\{\mathbf{p}_{i,1}..\mathbf{p}_{i,N_i}\}$ from the bivariate Gaussian distribution specified by $S_i$. For each generated point, its ground-truth cluster label is $i$.
- Randomly generate $K$ different $D \times D$ ground-truth covariance matrices $\{\Sigma_1, \Sigma_2 \cdots \Sigma_K\}$. It is required that the pairwise Wasserstein-2 distances should all be greater than 1.0. This is to make sure that observations of different clusters are statistically different.
- For each point $\mathbf{p}_{i,j}$, compute its Euclidean distance $d_{i,j}$ to the cluster center $\mathbf{p}_i$. For this point, generate a perturbed covariance matrix $P_{i,j}$ by adding Gaussian noise of scale $d_{i,j}\alpha$ to the ground-truth covariance matrix $\Sigma_i$. Sample a $D$-dimensional feature vector $\mathbf{f}_{i,j}$ from $P_{i,j}$. Similarly the maximum noise scale should be no larger than $10\%$ of the maximum entry in the ground-truth covariance matrix. Then the collection of all $(\mathbf{f}_{i,j}, \mathbf{p}_{i,j})$ makes the dataset.

---

[3]https://scikit-learn.org/stable/modules/generated/sklearn.metrics.adjusted_rand_score.html
[4]https://scikit-learn.org/stable/modules/generated/sklearn.metrics.adjusted_mutual_info_score.html

Choice of hyperparameters: Larger $\alpha$ makes the synthetic data noisier and cluster boundaries fuzzier. Larger $k$ makes model estimation more accurate and stable, thus better clustering results.

**Real-world Dataset**

(1) Pavement Dataset. This dataset is a sensor-based, originally univariate time series collected by experts. Car sensors collect data while driving on different pavements (cobblestone, dirt and flexible). There are in total 1055 successive, variable-length subsequences of accelerometer readings sampled at 100 Hz. Each subsequence has a label from the aforementioned three pavement types. We use the first 10 entries of each subsequence as its feature vector, and treat the truncated data as a 1055-long, 10-dimensional multivariate time series. Our task is to put subsequences of the same pavement labels into the same clusters.

The detailed information can be found at `https://timeseriesclassification.com/description.php?Dataset=AsphaltPavementType`.

(2) Vehicle Dataset. This is a multivariate time series dataset collected by tracking the working status of commercial vehicles (specifically, dumpers) using smart phones and published in the literature. The original paper is here: http://kth.diva-portal.org/smash/record.jsf?pid=diva2

(3) Gesture Dataset. This dataset records hand-movements as multivariate time series. Each movement record is 315-time-step long, and each time-step has a 3-dimensional vector, representing the spatial coordinates of the center of the hand. There are in total 2238 records, each record 315-time-step long, thus the entire length of the dataset is 704,970 time-steps. All the records belong to one of the eight gestures. We randomly shuffle the order of the records, so that it is more challenging. Our task is to cluster time-steps into different gestures. The detailed information about this dataset can be found in `https://timeseriesclassification.com/description.php?Dataset=UWaveGestureLibrary`.

(4) Climate Dataset. This dataset consists of locations on the earth and their 5 climate attributes (temperature, precipitation, wind, etc.). The ground-truth labels are the climate types of each location. There are in total 4741 locations, belonging to 14 different climate types. We use the great circle distance as the spatial distance metric for this dataset.

(5) iNaturalist-2018 Dataset. This dataset contains images of species from all over the world together with their geotags (longitude and latitude). The entire dataset is huge and geospatially highly imbalanced (e.g., there are in total 24343 images in the test set, but 10792 out of them are in the contiguous US). We use the ImageNet-pretrained Inception V3 model to embed each image into a 2048-dimensional vector as Mac Aodha et al. (2019); Mai et al. (2023b;a) did, and reduce it to a 16-dimensional vector using PCA, for the sake of computability of STICC. The ground-truth labels of each image are hierarchical (i.e., from the top kingdom types to the bottom class types), and we use the 6 kingdom types as the cluster labels. Dataset (4) gives an example of spatially-constrained clustering in the multivariate raw feature space, and Dataset (5) extends the boundary to the latent representation space of images.

(6) (7) For real-world dataset, we use the NYC Check-in data proposed by (Yang et al., 2015) and used in one of our baselines (Kang et al., 2022). This dataset contains check-in data in New York City by Foursquare, based on social media records. Each record includes VenueId, VenueCateg (POI Type), check-in timestamp (Weekday + Hour) and geospatial coordinate (Longitude + Latitude). We define the feature vector to be the normalized check-in vector, i.e., sum up the Hour attribute grouped by Week, and normalized this 7-dimensional vector. It is a feature vector representing the check-in patterns from Monday to Sunday. For evaluation, we construct 2 sets of ground-truth labels. One is from the NYC Check-in data itself: for each observation, we add up the one-hot POI type vectors of its nearest 50 neighbors and normalize it to be the POI embedding of this observation. Then, we cluster over these POI embeddings, and use the clustering labels as the ground-truth. Notice there is no information leak because our algorithm is fitted on check-in data and geo-coordinates only. The other is based on the Primary Land Use Tax Lot Output (PLUTO) dataset from NYC Open Data[5]. We extract the land-use records and assign to each observation the nearest land-use record as its ground-truth land-use label. For the sake of data quality, we only use the records of Manhattan and Bronx.

---

[5]https://data.cityofnewyork.us/City-Government/Primary-Land-Use-Tax-Lot-Output-PLUTO-/64uk-42ks/data

**Further Discussion on Experiment Results**

In Dataset (4) and Dataset (5) STICC/LMCC without spatial constraints yield lower performance than GMM, because the spatial sampling rate is too low (i.e., there are too few data points within a unit distance). There is a dilemma to STICC/LMCC algorithms: in order to obtain an adequate number of samples, we need to increase the sampling radius; however, as the sampling radius gets bigger, the samples become more noisy. In both cases the estimated distributions are inaccurate. Essentially, this problem originates from the balance between data sparsity (when having small neighborhood), and temporal/spatial incontiguity (when having large neighborhood) We address this problem by introducing a global prior. Since GMM can give a fairly good global estimation of the distribution of each cluster, we can use it as the prior distribution and update it in a maximum likelihood/Bayesian way given subsequence/subregion observations. This approach demonstrates a large increase in clustering performance for iNaturalist-2018. Again it demonstrates how important spatial metric information is in clustering data points that satisfy local metric constraints. This finding may lead to future works.

## 7.4 LMCC-MBC LOSS AS GOODNESS-OF-FIT TESTS

Existing model-based clustering algorithms formulate their clustering objective from the perspective of data likelihood. For example, in TICC and STICC, $d_m^2(\mathbf{f}_i, \mathbf{f}_j, \theta_i, \theta_j; \mathcal{M}) = -\log(\mathbb{P}(\mathbf{f}_i, \theta_j)) = -\log(\mathbb{P}(\mathbf{f}_i, \theta_{C_k})), j \in C_k$. Here the equality holds because of the assumption that observations in a cluster share a common underlying model, i.e. $\theta_j = \theta_{C_k}$ if $j \in C_k$. Although data likelihood as a loss function fits well with model selection criterions such as Akaike Information Criterion (AIC), it also leads to non-trivial top-down optimization procedures such as EM, which have several drawbacks: Firstly, the EM step is very time consuming and takes many iterations to converge. Secondly, the clustering result relies heavily on the initial assignment, and the optimization is subject to local optima. Thirdly, tuning the hyper-parameters (e.g., number of clusters) requires expensive retraining. These three problems can be avoided by using a bottom-up clustering algorithm, i.e., data points merge into clusters according to their underlying model similarity (like DBSCAN). Another issue with data likelihood objectives is the lack of flexibility when modeling constraints such as spatial metric autocorrelation as a generative process. For example, in TICC and STICC, the metric penalty function $r(i, j)$ is defined as $\mathbb{1}\{\tilde{X}_i \notin C_k\}$, where $\tilde{X}_i$ is the nearest neighbor of $X_i$. It solely depends on the metric distance, without considering correlations with their underlying models.

In order to enable efficient and flexible bottom-up clustering, we propose to formulate the loss function in terms of only pairwise computations between data points. Our solution mainly relies on goodness-of-fit tests (i.e., whether two samples come from a statistically identical distribution) as the pairwise computation. We punish the pairs that have large goodness-of-fit test statistics beyond a significance threshold with a hinge loss. This threshold is based on the mean goodness-of-fit test statistic, which potentially incorporates metric distance information. Specifically, the generic $\mathcal{L}^{\text{mcmb}}$ loss (Equation 4) can be implemented with goodness-of-fit tests as follows:

$$
\mathcal{L}^{\text{LMCC-MBC}}(\mathcal{C}) = \sum_{\{C_k \in \mathcal{C}\}} \sum_{\{i,j \in C_k\}} \left[ \lfloor d_m^2(i,j) - (\widehat{\mathbb{E}}_{i',j' \in \mathbb{N}} d_m^2(i',j') - \delta^0) \rfloor_+ \right.
$$
$$
\left. + \beta \lfloor d_m^2(i,j) - (\widehat{\mathbb{E}}_{i',j' \in \mathbb{N}_{i,j}} d_m^2(i',j') - \delta) \rfloor_+ \right]
\tag{16}
$$

Here $\mathbb{N}$ is the set of all example pairs, and $\mathbb{N}_{i,j} = N(d_c(i,j) \pm \epsilon)$ is the sample pairs in $(i,j)$'s distance bin as defined in semivariogram (Section 4). $\lfloor x \rfloor_+ = \max(0, x)$ is a rectifier (hinge) function. With a bit of overload of notations, we use $d_m^2(i,j)$ to represent a goodness-of-fit test statistic, such as the square of Wasserstein-2 distance (Panaretos & Zemel, 2019).

This proposed loss has the following properties:

- $d_m^2(i,j)$ is a goodness-of-fit test statistic. Thus, $d_m^2(i,j)$ being smaller than certain threshold implies data points $i$ and $j$ can pass the good-of-fit hypothesis test under confidence level.

- $\widehat{\mathbb{E}}_{i',j' \in \mathbb{N}} d_m^2(i,j) - \delta^0$ and $\widehat{\mathbb{E}}_{i',j' \in \mathbb{N}_{i,j}} d_m^2(i,j) - \delta$ can be seen as two thresholds based on the average test statistics plus a desired significance level. In the case of Wasserstein-2 distance (Panaretos & Zemel, 2019), the test statistics follows a normal distribution with the Wasserstein-2 distance of the true underlying models as the mean. We do not have access to these means, and

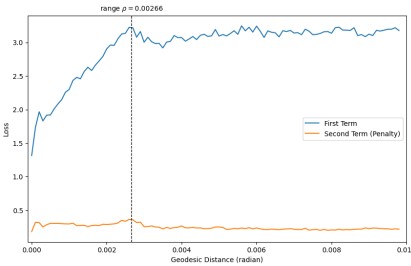

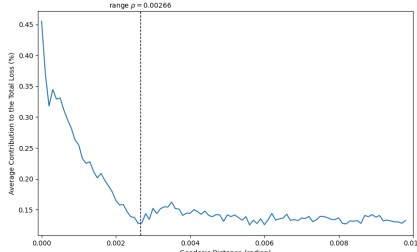

(a) The average value of the first term and the second term in Eq 16

(b) The average contribution of the second term in Eq 16 to the total loss.

Figure 3: Analysis of the loss composition in Eq 16. The average contribution of the metric-constraint penalty term to the total loss beyond the range quickly drops down to below $15\%$, which can be ignored in practice.

use empirical means in their places. While $\widehat{\mathbb{E}}_{i',j' \in \mathbb{N}} d_m^2(i,j) - \delta^0$ is independent to metric auto-correlation, $\widehat{\mathbb{E}}_{i',j' \in \mathbb{N}_{i,j}} d_m^2(i,j) - \delta$ is dependent. We call them *non-metric threshold* and *metric threshold* respectively. $\widehat{\mathbb{E}}_{i',j' \in \mathbb{N}_{i,j}} d_m^2(i,j)$ generalizes the classical semivariogram (Matheron, 1963), and we formally define it as Model-based semivariogram in Section 4.1.

- The rectifier (hinge) function avoid punishing example pairs which passes the test, and also avoids negative values in the sum. The idea is similar to the idea of margin and hinge loss in SVMs. Our ablation study (Appendix 7.4) shows that this choice increases computational stability and clustering accuracy.

In practice, since both $\widehat{\mathbb{E}}_{i',j' \in \mathbb{N}} d_m^2(i,j)$ and $\delta^0$ are constants, for computational efficiency we can simplify Equation 16 by defining $\delta^0 = \widehat{\mathbb{E}}_{i',j' \in \mathbb{N}} d_m^2(i,j)$ and $r(i,j) = \lfloor d_m^2(i,j) - (\widehat{\mathbb{E}}_{i',j' \in N_{i,j}} d_m^2(i,j) - \delta) \rfloor_+$, and rewrite our loss as

$$\mathcal{L}^{\text{LMCC-MBC}}(\mathcal{C}) = \sum_{\{C_k \in \mathcal{C}\}} \sum_{\{i,j \in C_k\}} [d_m^2(i,j) + \beta r(i,j)] \tag{17}$$

which is exactly Equation 4. This demonstrates that by choosing certain types of $d_m^2$ and $r(i,j)$, we can theoretically formulate bottom-up clustering as maximizing the intra-cluster pairs of data points that pass goodness-of-fit tests. This is the central formula that our algorithm is based on.

Finally, we condition the penalty function $r(i,j)$ on the range $\rho$ of the generalized model-based semivariogram (Equation 11). The reason being that when metric autocorrelation is absent ($d_m^2 > \rho$), the second term in Eq 16 is much smaller than the first term, and therefore can be ignored to save computation. Figure 3 shows an empirical analysis on the iNaturalist-2018 dataset. The average contribution of the metric-constraint penalty term to the total loss beyond the range quickly drops down to below $15\%$, which can be ignored in practice. In fact, we empirically verified in Appendix 7.7.4 that using the conditional form of $r(i,j)$ is both beneficial for improving clustering performance and for reducing computational expense.

## 7.5 THEORETICAL COMPLEXITY AND EMPIRICAL EXECUTION TIME

We denote $d$ as the data dimension, $n$ as the number of data points, and $K$ as the number of clusters. Theoretically, the complexity of LMCC is $O(n^2 d^2)$. Firstly, we need to estimate covariance matrices for each data point, which is $O(n^2 d \min(n, d))$. Since in most cases, $n >> d$, the complexity becomes $O(n^2 d^2)$. After estimating the covariances, we compute the pairwise Wasserstein-2 distances, which is again $O(n^2 d^2)$, because we need to do matrix multiplication ($O(d^2)$) $n^2$ times. Finally, we apply a distance-based clustering algorithm like DBSCAN on the Wasserstein-2 distance matrix, which is again $O(n^2)$. Thus the overall time complexity of LMCC is $O(n^2 d^2)$. This means, theoretically the execution time of TICC/STICC is $C \cdot K$ times of that of LMCC.

Table 2: Performance on 1-D and 2-D real-world datasets. $d$ denotes the feature dimension, $c$ denotes the ground-truth cluster number and $N$ denotes the size of each dataset. RT denotes the average run-time in seconds. **Bold** numbers and underlined numbers indicate the best and second best performances. TICC applies to 1-D datasets and STICC applies to 2-D datasets. $\beta_0$ means there is no temporal/spatial penalty term applied. NC means the algorithm does not converge. LMCC-wo/LMCC-w represents LMCC model without/with metric information respectively.

| Model | Temporal Dataset (1-D) | | | | | | | | | Spatial Dataset (2-D) | | | | | |
| | Pavement $d$=10, $c$=3 N=1,055 | | | Vehicle $d$=7, $c$=5 N=16,641 | | | Gesture $d$=3, $c$=8 N=704,970 | | | Climate $d$=5, $c$=14 N=4,741 | | | iNat2018 $d$=16, $c$=6 N=24,343 | | |
| | ARI | NMI | RT | ARI | NMI | RT | ARI | NMI | RT | ARI | NMI | RT | ARI | NMI | RT |
| GMM | 28.05 | 28.74 | < 1s | 57.87 | 58.78 | 3s | 2.44 | 4.15 | 14s | 19.06 | 34.97 | < 1s | 21.72 | 35.91 | 9s |
| (S)TICC-$\beta_0$ | 58.54 | 58.83 | 383s | 40.12 | 45.86 | 441s | 3.26 | 6.56 | 4782s | 13.30 | 30.53 | 1277s | NC | NC | 6881s |
| (S)TICC | 62.27 | 61.89 | 508s | 50.53 | 53.68 | 566s | 12.20 | 23.20 | 4511s | 17.62 | 37.29 | 1204s | NC | NC | 6325s |
| LMCC-wo | 76.10 | 74.36 | 14s | 63.31 | 58.60 | 74s | 8.12 | 33.60 | 573s | 16.63 | 36.73 | 746s | 21.90 | 36.47 | 588s |
| LMCC-w | **77.64** | **77.22** | 14s | **65.04** | **59.36** | 76s | **26.51** | **55.34** | 554s | **20.08** | **40.91** | 755s | **42.70** | **40.49** | 594s |

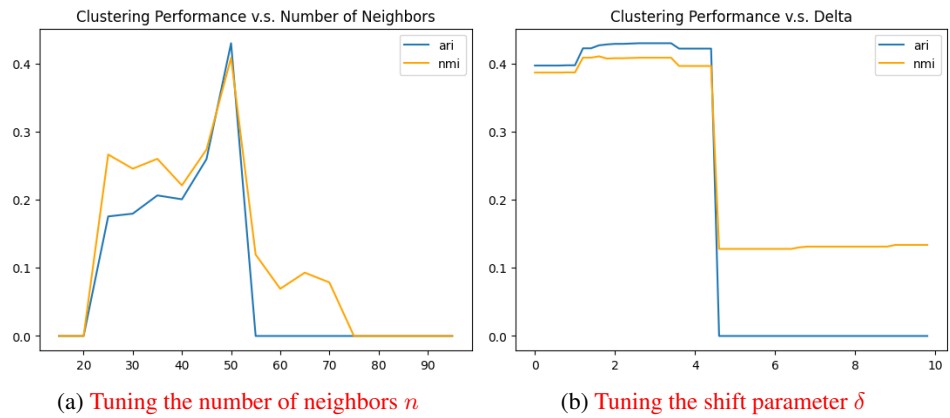

(a) Tuning the number of neighbors $n$      (b) Tuning the shift parameter $\delta$

Figure 4: The performance curve with regard to the grid-searched hyperparameters $n$ and $\delta$

Next, we show that the time complexity of the SOTA models (TICC and STICC) is $O(C \cdot K \cdot n^2 d^2)$, where $C$ is how many iterations it takes to converge, which usually increases as $K$ and $n$ increase.

TICC/STICC needs to 1) compute an initial cluster assignment by kMeans, which is $O(n^2)$; 2) estimate cluster-wise covariance matrices and compute the likelihood of each data point against each cluster, which is $O(K \cdot n^2 d^2)$; 3) update cluster assignment, which is reported $O(K \cdot n)$ in the original papers; 4) repeat (1) to (3) $C$ times until convergence. Thus the overall time complexity is $O(C \cdot K \cdot n^2 d^2)$.

We also evaluated the empirical time complexity of each clustering algorithm. Please refer to the "RT" column in Table 2. We can see that TICC/STICC is much slower than our LMCC. Notice the time TICC/STICC takes highly depends on how many iterations it takes to converge.

Finally, the spatial complexity of both LMCC and TICC/STICC is $O(n \cdot d^2)$, since all we need to store is the covariance matrices of each data point.

## 7.6 HYPERPARAMETER TUNING

We include an ablation study to investigate the influence of shift hyperparameter and the number of neighbors using the most complicated iNaturalist 2018 dataset. Figure 4 demonstrates that the

Table 3: Clustering Performance using Different Covariance Estimation Methods

| Method | TICC (Baseline) | | LMCC-MBC (GLasso) | | LMCC-MBC (MinCov) | | LMCC-MBC (Shrunk) | |
|---|---|---|---|---|---|---|---|---|
| | ARI | NMI | ARI | NMI | ARI | NMI | ARI | NMI |
| Performance | 62.27 | 61.89 | 77.64 | 77.22 | 80.82 | 73.78 | 74.70 | 71.42 |

Table 4: Clustering Performance using Different Distance-based Clustering Algorithm

| Method | TICC (Baseline) | | LMCC-MBC (DBSCAN) | | LMCC-MBC (HDBSCAN) | | LMCC-MBC (OPTICS) | |
|---|---|---|---|---|---|---|---|---|
| | ARI | NMI | ARI | NMI | ARI | NMI | ARI | NMI |
| Performance | 62.27 | 61.89 | 77.64 | 77.22 | 72.35 | 69.61 | 69.77 | 68.58 |

search space of single hyperparameters has good convexity. Thus, we can easily and quickly tune the hyperparameters by hierarchical grid search.

For tuning $\delta$, we do not need to re-compute the covariance matrices. Instead, we only need to re-run the distance-based clustering algorithm like DBSCAN. Thus the time complexity of a complete grid search is only $O(A \cdot Bn^2)$, where A and B are the grid sizes of the number-of-neighbor hyperparameter and the shift hyperparameter.

Instead, the competing baselines TICC/STICC must re-run the entire algorithm when tuning hyperparameters. That means the complete grid search is $O(A \cdot B \cdot C \cdot K \cdot n^2 d^2)$, even if we only tune the most important $\lambda$ and $\beta$ hyperparameters.

## 7.7 ABLATION STUDIES

We conduct a series of ablation studies on the pavement dataset to investigate the effectiveness of different model component choices.

### 7.7.1 REPLACING GRAPHICAL LASSO WITH OTHER COVARIANCE ESTIMATION ALGORITHMS

We used Graphical Lasso in our paper as the covariance estimation algorithm, but the effectiveness of LMCC-MBC does not rely on this specific implementation. As an ablation study, we replace Graphical Lasso with Minimum Covariance Determinant (MinCov)[6] and Shrunk Covariance (Shrunk)[7] and apply LMCC-MBC on the Pavement dataset. Clustering performance is reported in Table 7.4. All results are under the best hyperparameters after grid-search of hyper-parameters.

In general, the more robust and more accurate the covariance estimation algorithm is, the better the clustering performance is. Shrunk is the least robust covariance estimation algorithm among the three, thus its performance is obviously lower than GLasso and MinCov. However, different variations of LMCC-MBC still significantly outperform the strongest baseline, TICC. It indicates the effectiveness of LMCC-MBC.

### 7.7.2 REPLACING DBSCAN WITH OTHER DISTANCE-BASED CLUSTERING ALGORITHMS

Similarly, LMCC-MBC does not rely on any specific implementation of the clustering algorithm. Table 7.4 again demonstrates that though clustering performances are affected by the choice of distance-based clustering algorithms, LMCC-MBC still outperforms the baselines by large margins.

### 7.7.3 REMOVING THE RECTIFIER (HINGE) OPERATION IN THE LOSS FUNCTION

---

[6]https://scikit-learn.org/stable/modules/generated/sklearn.covariance.MinCovDet.html
[7]https://scikit-learn.org/stable/modules/generated/sklearn.covariance.ShrunkCovariance.html

Table 5: Clustering Performance using Conditional and Unconditional Penalty Terms

| Method | LMCC (Conditional) | | LMCC-MBC (Unconditional) | |
|---|---|---|---|---|
| | ARI | NMI | ARI | NMI |
| Performance | 77.64 | 77.22 | 76.91 | 76.25 |

In our ablation experiment, removing the rectifier (hinge) operation causes computational instability. Since we apply a distance-based clustering algorithm on top of the weighted distance matrix, all entries are required to be positive. When we remove the maximum operation, the weighted distance sometimes becomes negative and the clustering algorithm fails.

### 7.7.4 REMOVING THE RANGE CONDITION IN THE LMCC PENALTY FUNCTION

As an ablation study, we ignore the range condition in Equation 11, i.e., define $r(i, j)$ simply as

$$r(i, j) = \max(0, d_m^2(i, j) - \gamma_m(d_c(i, j)) + \delta) \tag{18}$$

This means we need to compute the penalty term for all possible pairs of data points. We then apply LMCC-MBC. The comparison of clustering performance using Equation 11 (Conditional) and using Equation 18 (Unconditional) is demonstrated in Table 7.4. Ignoring the range does not only negatively affects the clustering performance, but also wastes resources, since we can spare the computation of penalty terms of pairs out of range.

