# OpenReview forum: "LMCC-MBC: Metric-Constrained Model-Based Clustering with Wasserstein-2 Distance of Gaussian Markov Random Fields"
_ICLR.cc/2024/Conference — Submitted to ICLR 2024_

### Official Review · Reviewer_cTE9 · 2023-10-26

**Soundness:** 3 good
**Presentation:** 2 fair
**Contribution:** 3 good
**Rating:** 6
**Confidence:** 5

**Summary:**

Problem: This paper studies a clustering problem for data with special properties like time or spatial positions. This paper focuses on the case when there is the effect of metric autocorrelation in data, which means the variance of feature vectors is positively correlated to their temporal/spatial distances.

Modelling: The authors propose a metric-constrained model-based model that leverages the correlation between adjacent estimated models and their locations in metric space. They use Gaussian Markov Random Fields to model inter-observation dependency and use Wasserstein-2 distance to measure the distance between estimated model parameters. Because of the metric autocorrelation, they use a metric penalty that decreases as distance increases.

Key results: From the experimental results, their algorithm appears to be computationally more efficient than other methods and has better performance in terms of ARI and NMI.

**Strengths:**

1. From the experimental results, their algorithm appears to be computationally more efficient than other methods and has better performance in terms of ARI and NMI.
2. The proposed method can deal with arbitrary dimensions of constraint space instead of just 1-D and 2-D.

**Weaknesses:**

Major comments:
1. In the contribution section: the authors indicate that they present a solid mathematical proof of the soundness of their generalized semivariogram. But there isn’t any proof in this paper and later they say “The soundness of this generalized definition can be easily verified on real-world datasets”.
2. Same problem for the fourth contribution. It would be better to at least have a complexity analysis for your algorithm to say that your algorithm solves the problem efficiently with “solid theoretical proofs”.
3. I think the authors need to justify the use of GMRF in the model since they are fitting a GMRF for each sample. It would make more sense to fit one GMRF for one cluster as different data clusters will naturally have different variable dependency structures.
4. One of the advantages of the proposed method they can deal with arbitrary dimensions of constraint space instead of just 1-D and 2-D. Did you try it with a 3-D dataset?

Minor comments:
1. In section 5.1, the authors claim that the proposed method has only three hyperparameters to tune. What about the tuning parameter for Graphical Lasso?

**Questions:**

See "Weaknesses"

---

> ### Author Response · Authors · 2023-11-22
> **Response to Weaknesses/Questions**
>
> **Q1: solid mathematical proof of the soundness of their generalized semivariograms**
>
> **A1:** Briefly speaking, the expected Wasserstein-2 distance for a given metric distance bin is part of the computation for determining the threshold of goodness-of-fit tests, and we formally define it as Model-based semivariogram in Section 4.1 since it has similar form to the classical semivariograms. A detailed discussion and derivation can be found in Appendix 7.4. *
>
> **Q2: a complexity analysis for LMCC algorithm**
>
> **A2:** Please kindly refer to our general response Q3 as well as Appendix 7.4 for a detailed discussion and additional experiments.
>
> **Q3: justify the use of GMRF for each sample rather than learning a GMRF for one cluster**
>
> **A3:** Please see Part 2 of our general response as well as Appendix 7.4 about the main contribution of our work: a novel family of clustering objectives -- i.e., instead of optimizing the total data likelihood, we propose to minimize the number of failed goodness-of-fit tests, which compare pairs of data points and their underlying models in the clustering.
>
> Fitting one model (e.g., GMRF) for one clustering is commonly used in existing model-based clustering algorithms such as TICC and STICC, which optimize the total data likelihood. They require a non-trivial  estimation-maximization (EM) procedure which leads to several drawbacks in efficiency and modeling:.
> Firstly, the EM step is very time-consuming and takes many iterations to converge. Secondly, the clustering result relies heavily on the initial assignment, and the optimization is subject to local optima. Thirdly, tuning the hyper-parameters (e.g., number of clusters) requires expensive retraining. Finally, data likelihood objectives have a lack of flexibility when modeling constraints such as spatial metric autocorrelation as a generative process. These problems can be avoided by using a bottom-up model-based clustering algorithm, i.e., data points merge into clusters according to their underlying model similarity (like DBSCAN).
>
> In order to enable a bottom-up model-based clustering, we need to learn a GMRF for each data sample instead of learning one GMRF for one cluster. This allows us to perform the goodness-of-fit test (i.e., test whether two samples come from a statistically identical distribution) between any given two data samples in order to determine whether we can merge these two data samples into one cluster.
>
> In short, we estimate one GMRF for one data sample instead of one cluster to enable a  bottom-up DBSCAN-like model-based clustering rather than the traditional kMeans-like practice.
>
> **Q4: Apply LMCC on 3D dataset**
>
> **A4:** This is a very good point. We have found 3-D datasets about point cloud-based scene segmentation and are working on the experiments. Due to time limits of rebuttal, we can not present the results in the rebuttal, but we will add this result to the camera-ready version.
>
>
> **Q5: Tuning parameter for Graphical Lasso**
>
> **A5:** We grid-searched the hyperparameters alpha, tolerance and maximum number of iterations for Graphical Lasso. The hyperaramters we use are alpha=0.01, tolerance=1e-4 and maximum iterartion=1000. Theoretically, as long as Graphical Lasso converges, it yields very stable covariance estimation. Thus there is no need to tune these hyperparameters on the entire LMCC-MBC algorithm. We simply find out hyperparameters that enables Graphical Lasso to converge on a subset of the dataset and fix them throughout the experiments.

---

> > ### Comment · Reviewer_cTE9 · 2023-11-23
> >
> > I appreciate the authors' responses, which have addressed most of my concerns.
> >
> > As for Q3, I understand your choice of GMRF for performming the goodness-of-fit test. But it is still hard for me to accept that a GMRF learned from a single sample would have sufficient statistical significance for such tests.
> >
> > Overall, I would like to keep my score.

---

> > > ### Author Response · Authors · 2023-11-23
> > >
> > > Thank you for your response!
> > >
> > > For Q3, the GMRF is learned from the neighborhood of a sample, i.e., what the hyperparameter $n$ of "number of neighbors" controls. We usually choose $n>30$ so that the estimation is statistical significant. This is a common practice in model-based clustering. For example, TICC and STICC both used the neighborhood to estimate underlying models for each sample. The difference is that they additionally estimate a common underlying model for each cluster.
> > >
> > > I wish this addresses your concerns.

---

### Official Review · Reviewer_FyBz · 2023-10-30

**Soundness:** 2 fair
**Presentation:** 3 good
**Contribution:** 2 fair
**Rating:** 6
**Confidence:** 3

**Summary:**

In this paper, the authors focus on metric-constrained clustering (when clustering is based not only on the features of the data points but also on constraints in a metric space (time, geo data points). Within that framework, they propose a new metric-constrained model-based clustering approach, LMCC-MBC (Locally Monotonically and Continuously Constrained Model-Based Clustering) optimized to maximize intra-cluster cohesion and inter-cluster separation, working as follows:

-For each data point, compute the neighboring set of points in the metric-constraint space (not using the features) and fit a Gaussian Markov Random Field model with Graphical Lasso algorithm.

-For each pair of observations, compute model and metric distances (at this stage, we still don’t leverage features of the data points).

-Compute a semivariogram from the two distances and $\rho$ the range of the fitted semivariogram

-Compute weighted distance matrix M based on model, metric distances, semivariogram, $\rho$

-Run the some density-based clustering method (ex: DBSCAN) on M for the final clustering partition.

Wasserstein-2 distance is used as the model distance.

**Strengths:**

The paper is well written and quite enjoyable to read.

The only hyperparameters of the clustering approach are the number of neighbors to consider per data point, the metric-constraint strength ($\beta$) and a shift parameter  ($\delta$) that was found empirically to overlap the clustering boundaries if appropriately tuned.

The authors generalize the concept of the classic semivariogram for multivariate data points.

Experiments are made on 2 synthetic datasets and 7 real-world datasets equally split between the temporal and spatial use case. LMCC model with and without metric constraints show promising results by providing systematically the best ARI and NMI.

A comparison is made between LMCC and TICC/STICC algorithms (the competing approaches for the metric-constrained case) regarding stability and robustness.

**Weaknesses:**

As with many clustering algorithms, there could be a theoretical comparison analysis of the space and time complexities for LMCC-MBC and competing SOTA techniques.

Regarding the following claim:  “In fact, Wasserstein-2 distance is the only feasible choice of model distance that theoretically guarantees the generalized model-based semivariogram is compatible with the classic definition. Consequently, GMRF is chosen because it is the most computationally efficient model parameterization under Wasserstein-2 distance. The following section will prove this in detail.”
It does not sound to me that there is a proof here. The usage of Wasserstein-2 distance and GMRF is justified, yes, but I don't have the feeling that this proves that these are the only options as stated.

No experimental study of the effects of the number of neighbors and the shift parameter on accuracy.

Minor, typos:

-missing space after “unknown” in intro p.1

-requirment, p.7

-natrually, p.7 before eq. 11

-fittiing p.8 in Algorithm 1

-hyperparamters p.16

-missing space after “our baselines(Kang et al, 2022) p.16

-unlined instead of underlined in p.17

-It seems there is a missing paragraph in the Appendix related to Execution time comparison

**Questions:**

Q1: Could you please provide a theoretical comparison analysis of the space and time complexities for LMCC-MBC and competing SOTA techniques?

Q2: How in practice do you tune the shift parameter $\delta$? It seems to be the determinant hyperparameter of the method but there is no study to show the influence on accuracy. Same for the number of neighbors to consider.

Q3: Can you please explain how you prove that the requirement of the Wasserstein-2 distance between estimated model parameters from GMRF (model distance) to be a locally monotonic continuous function of the metric distance guarantees intra-cluster cohesion and inter-cluster separation?

Q4: What is the purpose of the experiments with the synthetic datasets? They seem to be applying the same experiment setup as the real-word ones.

=== AFTER REBUTTAL ===

I thank the authors for taking the time to answer my questions that are now addressed (time complexity analysis, further justification of the Wasserstein-2 distance). Hence, I upgrade my score to Weak accept.

---

> ### Author Response · Authors · 2023-11-22
> **Response to Weaknesses/Questions**
>
> **Q1: Theoretical comparison analysis of the space and time complexities for LMCC-MBC and competing SOTA techniques**
>
> **A1:** Please refer to our general response Q3 as well as Appendix 7.5 for a detailed discussion and additional experiments.
>
> **Q2: No experimental study of the effects of the number of neighbors and the shift parameter on accuracy.**
>
> **A2:** Please refer to our general response Q4 and Appendix 7.6 for the added ablation studies on these two hyperparameters.
>
> **Q3: How to prove that LMCC guarantees intra-cluster cohesion and inter-cluster separation**
>
> **A3:** Thank you for this question! Please refer to Appendix 7.4, where we carefully discussed how the LMCC-MBC loss can be seen as a weighted sum of penalties that punish intra-cluster data pairs for not passing a goodness-of-fit test. Consequently, minimizing this loss equals encouraging the intra-cluster data points to be as similar as possible, i.e., intra-cluster cohesion in a model-based clustering sense. Since different clusters have different underlying models, increasing intra-cluster cohesion will automatically increase inter-cluster separation.
>
> **Q4: The proof of the usage of Wasserstein-2 distance and GMRF**
>
> **A4:** Please refer to our general response Q2 for the reason for our choice of Wasserstein-2 distance, GMRF, Graphical Lasso, and DBSCAN. We also provide the theoretical foundation of the reason why we use Wasserstein-2 distance (see our general response Q1). Essentially, in order to enable a bottom-up model-based clustering, we need to formulate the loss function (see Equation 4 ) in terms of only pairwise computations between data points. The goodness-of-fit test (i.e., test if two samples come from a statistically identical distribution) statistic perfectly matches our needs. The square of Wasserstein-2 distance (Panaretos & Zemel, 2019) satisfies our requirements since it metricizes weak convergence.
>
> **Q5: the purpose of the experiments with the synthetic datasets?**
>
> **A5:** The purpose of experiments on the synthetic dataset is to provide an understanding of whether our theory is solid, since the real-world datasets may or may not follow all the theoretical assumptions we make while we can control the data generation process of the synthetic dataset and make it follow our assumptions. It can be seen that on synthetic datasets, as all the assumptions are met, the performance of our method is extremely high (~90%), while in real-world datasets, because of violations of assumptions (e.g., data error, non-normality, etc.) the performance may drop but still significantly outperform the existing STOA clustering algorithms.
>
> **Q6: Typos.**
> **A6:** We are sorry about these typos. We have corrected the typos in our updated version, highlighted in red.

---

### Official Review · Reviewer_FzKt · 2023-10-31

**Soundness:** 2 fair
**Presentation:** 3 good
**Contribution:** 1 poor
**Rating:** 5
**Confidence:** 2

**Summary:**

This work attempts to address the metric autocorrelation problem in model-based clustering. To be specific, each data sample is modeled by a Gaussian Markov Random Field, and the distance between GMRFs are measured by Wasserstein-2 distance. The conventional clustering assumption objective is then optimized to minimize intra-cluster distances and maximize inter-cluster distances. The authors argue that the combination of Wasserstein-2 distance and GMRF is cautiously chosen and provide some theoretical analyses.

**Strengths:**

* The paper is well-written and easy to read.
* This work proposes to incorporate metric autocorrelation into the clustering model, which is important but overlooked by previous works.
* Empirical results on both synthetic and real-world datasets verify the effectiveness of the proposed method.

**Weaknesses:**

* The effectiveness of modification from Eq. (5) to (6) is empirical and lacks a theoretical guarantee.
* The proposed method just combines several existing components. Even though the authors argue that the chosen combination is not heuristic, and provides some special theoretical properties of them, the overall technical contribution looks less significant to me.
* The experiments are weak. Only an overview of clustering performance is provided. Ablation studies of the design choices are lacking, so these claims are not well-supported. Comparisons with strong baselines are also lacking.

------

## Post-rebuttal

Dear authors, I greatly appreciate the detailed clarification. Some of my concerns are cleared.

Regarding A1 & A2, the major contribution is minimizing the number of failed goodness-of-fit tests rather than optimizing the total data likelihood according to General Response Q1, which seems novel. However, the components are still taken as-is despite being well-crafted as explained in General Response Q2, so my original comment still holds.

Regarding A4, I don't work in your area so I can't suggest stronger baselines. But from a general perspective, TICC was published in 2017 hence doesn't look new. STICC was published in a geoscience journal rather than ML/DL journals/conferences, and it has been cited only 7 times as of now, that's why I had the question.

Overall, I raised my rating to 5.

**Questions:**

N/A

---

> ### Author Response · Authors · 2023-11-22
> **Response to Weaknesses/Questions**
>
> **Q1: The effectiveness of modification from Eq. (5) to (6) is empirical and lacks a theoretical guarantee**
>
> **A1:** Thank you for raising this important question! We provide more detailed reasoning in the general rebuttal Q1 and Appendix 7.4. Briefly speaking, the expected Wasserstein-2 distance for a given metric distance bin is part of the computation for determining the threshold of goodness-of-fit tests, and we formally define it as Model-based semivariogram in Section 4.1 since it has similar form to the classical semivariograms.
>
> **Q2: ​​The proposed method just combines several existing components. The overall technical contribution looks less significant to me.**
>
> **A2:** Thank you for  raising this important question regarding our work! Please kindly refer to the general rebuttal Q2 for a discussion about our main contribution, which we briefly summarize here:
> The most important contribution of our work  is  a novel family of clustering objectives -- i.e., instead of optimizing the total data likelihood, we propose to minimize the number of failed goodness-of-fit tests, which compare pairs of data points and their underlying models in the clustering. Compared to total data likelihood based objectives, this novel approach enjoys several benefits in efficiency and modeling (see details in Appendix 7.4).
>
> The choice of specific implementations, i.e. Graphical Lasso and DBSCAN, is not important. Using other covariance estimation and distance-based clustering algorithms yields comparable results. We refer to Appendix 7.7.1 (Table 3) and 7.7.2 (Table 4) for the ablation studies on this. That demonstrates we are not mere heuristic combinations of existing algorithms.
>
> **Q3: Ablation studies of the design choices are lacking.**
>
> **A3:** Thanks for your suggestion. We updated our paper and added an ablation study on the influence of hyperparameters. Please kindly refer to the general rebuttal Q4 and Appendix 7.6. In general, the conclusion is that both the number-of-neighbor hyperparameter and the shift hyperparameter have nearly convex responses. That means we can easily tune them by simple hierarchical grid search.
>
> Additionally, Table 1 in our paper does not only report the overall clustering performance, but also compares the effectiveness of introducing spatial information (i.e., we compared performances w/wo spatial distance). We can see from the results that introducing metric distance into the clustering objective significantly improves the clustering accuracy.
> We also added ablations studies on the choice of covariance matrix estimation methods (*Graphical Lasso* v.s. *Minimum Covariance Determinant* v.s. *Shrunk Covariance*) and the choice of distance-based clustering (*DBSCAN* v.s. *HDBSCAN* v.s. *OPTICS*). Please refer to our general response Q2.3, Appendix 7.7.1 and Appendix 7.7.2.
>
> **Q4: Comparisons with strong baselines is lacking.**
> **A4:** To the best of our knowledge, in temporal and spatial clustering, TICC and STICC are the strongest baselines for metric constrained clustering. If  you can point us to other baselines we are happy to do more experimental comparisons.

---

### Author Response · Authors · 2023-11-22
**General Response Q1: Theoretical soundness of LMCC-MBC and the generalized model-based semivariogram**

**Q1: Theoretical soundness of LMCC-MBC and the generalized model-based semivariogram**

**A1, Part 1: Interpreting LMCC-MBC loss as goodness-of-fit tests**

We wish to clarify the soundness of our design of the LMCC-MBC loss function as well as the theoretical interpretation of our proposed generalized model-based semivariogram from the perspective of goodness-of-fit tests. LMCC-MBC is effectively minimizing the hinge loss penalty of data point pairs that do not pass the goodness-of-fit test (see Appendix 7.4, Eq 16). The loss combines two types of tests -- those independent to metric autocorrelation and those dependent to metric autocorrelation. Please see a detailed discussion in Appendix 7.4. Here we present a brief summary:

Wasserstein distance is commonly used for carrying out goodness-of-fit tests (Panaretos&Zemel 2018):  Given the empirical distribution $µ_n$ associated with the sample $X_1, . . . ,X_n ∼  µ$ and $Y_1, . . . , Y_m ∼ ν$ with corresponding empirical distribution $ν_m$, the squared Wasserstein-2 distance $d_m^2 = W_2^2(µ_n, ν_m)$ is a test statistic for the null hypothesis $µ = ν$. It follows a normal distribution (Panaretos&Zemel 2018, Eq 6) with the  Wasserstein-2 distance of the true underlying models as the mean. We do not have access to these means, and use empirical means (both with and without metric autocorrelation) in their places. The expected Wasserstein-2 distance for a given metric distance bin is part of the goodness-of-fit tests, and we formally define it as Model-based semivariogram in Section 4.1 since it has similar form to the classical semivariograms.

A downstream bottom-up clustering algorithm (e.g. DBSCAN) tries to find the clustering with the best sum of goodness-of-fit test value. This can be interpreted as finding clustering assignments that maximize the total significance of both significance tests.

**References:**

Statistical Aspects of Wasserstein Distances (Panaretos&Zemel 2018)

**A1, Part 2: Main contribution of our work -- Goodness-of-fit test vs likelihood-based loss functions**

The most important contribution of our work  is  a novel family of clustering objectives -- i.e., instead of optimizing the total data likelihood, we propose to minimize the number of failed goodness-of-fit tests, which compare pairs of data points and their underlying models in the clustering. The goodness-of-fit tests can integrate metric autocorrelation as part of the mean statistics. Please see detailed discussions in Appendix 7.4, but here we provide a brief summary:

Existing model-based clustering algorithms formulate their clustering objective from the perspective of data likelihood -- for example, in TICC and STICC. Although data likelihood as a loss function fits well with model selection criteria such as the Akaike Information Criterion (AIC), it also leads to non-trivial top-down optimization procedures such as EM, which have several drawbacks: Firstly, the EM step is very time-consuming and takes many iterations to converge. Secondly, the clustering result relies heavily on the initial assignment, and the optimization is subject to local optima. Thirdly, tuning the hyper-parameters (e.g., number of clusters) requires expensive retraining. Finally, the data likelihood objectives have a lack of flexibility when modeling constraints such as spatial metric autocorrelation as a generative process.

In order to enable efficient and flexible bottom-up clustering, we propose to formulate the loss function in terms of only pairwise computations between data points.  Our solution mainly relies on goodness-of-fit tests (i.e., whether two samples come from a statistically identical distribution) as the pairwise computation.  Please see a detailed discussion in Appendix 7.4.

**References:**

https://en.wikipedia.org/wiki/Akaike_information_criterion

---

### Author Response · Authors · 2023-11-22
**General Response Q2: The choice of Wasserstein-2 distance, GMRF, and other components**

**Q2: The choice of Wasserstein-2 distance, GMRF, and other components**

**A2:** We wish to summarize the reasons we choose Wasserstein-2 distance and GMRF here.

*1) The choice of  Wasserstein-2 distance*

As is stated in Part 1 of the general rebuttal, theoretically we need a goodness-of-fit statistic to investigate two random samples and test whether their underlying models are statistically identical. Wasserstein-2 distance is one of the statistics. The reason why Wasserstein-2 is unique is that it is the only computable statistical distance that metricizes weak convergence (Gibbs & Su, 2002). That means, even though other statistics such as KL-divergence can also be used to quantify how similar two distributions are, only Wasserstein-2 distance guarantees that when the distance becomes 0, two distributions are the same. This is a stronger encouragement for intra-cluster cohesion.

*2) The choice of GMRF*

The choice of the model is task-specific, and Wasserstein-2 distance can be applied to other types of models. The GMRF used in our study enjoys computational efficiency. While there are approximate algorithms that compute Wasserstein-2 distance between two arbitrary distributions, the Wasserstein-2 distance between 2 normal distributions has a closed form and can be very easily computed. LMCC can be easily applied to non-GMRF problems by changing the way we compute Wasserstein-2 distance. We will explore non-GMRF LMCC-MBC in future works.

*3) The choices of other components such as Graphical Lasso and DBSCAN*

The estimation of the covariance matrices (of GMRF) for each data point (e.g., Graphical Lasso) and the distance-based clustering (e.g., DBSCAN) afterward, do not rely on the choice of specific implementations. For example, Graphical Lasso can be replaced with *Minimum Covariance Determinant (MinCov)* or *Shrunk Covariance (Shrunk)*,  and DBSCAN can be replaced with *HDBSCAN* or *OPTICS*. In Appendix 7.7.1 and 7.7.2, we added ablation studies that compare the clustering performances of different LMCC variances by using different covariance matrix estimation algorithms and different distance-based clustering algorithms on the Pavement dataset. Please refer to Table 3 and Table 4 for the experimental results (we will add experiments for all datasets for the final version).
The result shows that though using different implementations of covariance matrix estimation and distance-based clustering affects the clustering performance, all variations of LMCC-MBC still significantly outperform the strongest baselines. For example,  LMCC with MinCov achieves 80.82 ARI and 73.78 NMI on the Pavement dataset compared to 77.64 ARI and 77.22 NMI with Graphical Lasso, while the baseline TICC only achieves 62.27 ARI and 61.89 NMI.
The analysis above justifies both theoretically and empirically that LMCC-MBC is a generic clustering framework and not a simple combination of existing components.
The complete ablation experiment results can be found in Table 3, Appendix 7.7.

---

### Author Response · Authors · 2023-11-22
**General Response Q3: Theoretical and Empirical Space-and-Time Complexity Analysis**

**Q3: Theoretical and Empirical Space-and-Time Complexity Analysis**

**A3:** First we show that the overall time complexity of LMCC is $O(n^2d^2)$:

Here, $d$ is the data dimension, $n$ is the number of data points, and $K$ is the cluster number. Firstly, we need to estimate covariance matrices for each data point, which is $O(n^2\cdot d\cdot min(n,d))$. Since in most cases, $n >> d$, the complexity becomes $O(n^2d^2)$. After estimating the covariances, we compute the pairwise Wasserstein-2 distances, which is again $O(n^2d^2)$, because we need to do matrix multiplication ($O(d^2)$) $n^2$ times. Finally, we apply a distance-based clustering algorithm like DBSCAN on the Wasserstein-2 distance matrix, which is again $O(n^2)$. Thus the overall time complexity of LMCC is $O(n^2d^2)$.

Next, we show that the complexities of the SOTA models (TICC and STICC) are $O(C\cdot K\cdot n^2d^2)$, where $C$ is how many iterations it takes to converge. $C$ usually increases with $K$ and $n$. This is because as $K$ and $n$ increase, it is harder to stabilize the cluster assignment.

TICC/STICC needs to 1) compute an initial cluster assignment by kMeans, which is $O(n^2)$; 2) estimate cluster-wise covariance matrices and compute the likelihood of each data point against each cluster, which is $O(K\cdot n^2d^2)$; 3) update cluster assignment, which is reported $O(K\cdot n)$ in the original papers; 4) repeat (1) to (3) C times until convergence. Thus the overall time complexity is $O(CK\cdot n^2d^2)$. This means, theoretically the execution time of TICC/STICC is $C\cdot K$ times longer than that of LMCC-MBC.

We also evaluated the empirical time complexity of each algorithm. Please refer to the “RT” column in Table 2 of Appendix 7.5. We can see that TICC/STICC is much slower than our LMCC-MBC. Notice the time TICC/STICC takes highly depends on how many iterations it takes to converge.

The spatial complexity of both LMCC and TICC/STICC is $O(n\cdot d^2)$, since all we need to store is the covariance matrices of each data point.

We have added these as an additional section in Appendix 7.5.

---

### Author Response · Authors · 2023-11-22
**General Response Q4: Ablation study on tuning hyperparameters**

**Q4: Ablation study on tuning hyperparameters**

**A4:** We added an ablation study to investigate the influence of shift hyperparameter and the number of neighbors in Appendix 7.6 (Figure 4) using the most complicated iNaturalist 2018 dataset. We show that the search space of single hyperparameters is nearly convex. Thus, we can easily and quickly tune the hyperparameters by hierarchical grid search.

For every hyperparameter choice, we do not need to re-compute the covariance matrices. Instead, we only need to re-run the distance-based clustering algorithm. Thus the time complexity of a complete grid search is only $O(ABn^2)$, where $A$ and $B$ are the grid sizes of the number-of-neighbor hyperparameter and the shift hyperparameter.

Instead, the competing baselines TICC/STICC must re-run the entire algorithm when tuning hyperparameters. That means the complete grid search is $O(ABCKn^2d^2)$, even if we only tune the most important \lambda and \beta hyperparameters.

---

### Meta-Review · Area_Chair_DeWU · 2023-12-04

**Metareview:**

This paper advocates for using the Wassestein-2 distance to measure discrepancy between GMRFs in a model-based clustering framework. It includes some nice technical novelty but ultimately the reviewers are lukewarm on the extent of the contributions. Several reviewers appreciated the thoughtful rebuttal but even after slightly adjusting their scores upward, the paper falls below the bar and can be improved by incorporating some of these constructive criticisms

**Justification For Why Not Higher Score:**

borderline reviews

**Justification For Why Not Lower Score:**

n.a

---

### Decision · Program_Chairs · 2024-01-16

Reject